# Quantifying ethnic segregation in cities through random walks

Sandro Sousa [1,2] ✉ & Vincenzo Nicosia [1] ✉

Socioeconomic segregation has an important role in the emergence of large-scale inequalities in urban areas. Most of the available measures of spatial segregation depend on the scale and size of the system under study, or neglect large-scale spatial correlations, or rely on ad-hoc parameters, making it hard to compare different systems on equal grounds. We propose here a family of non-parametric measures for spatial distributions, based on the statistics of the trajectories of random walks on graphs associated to a spatial system. These quantities provide a consistent estimation of segregation in synthetic spatial patterns, and we use them to analyse the ethnic segregation of metropolitan areas in the US and the UK. We show that the spatial diversity of ethnic distributions, as measured through diffusion on graphs, allow us to compare the ethnic segregation of urban areas having different size, shape, or peculiar microscopic characteristics, and exhibits a strong association with socio-economic deprivation.

Spatial heterogeneity is a characteristic aspect of a variety of complex systems, from urban areas to ecosystems[1,2], and the presence of non-trivial spatial patterns in the organisation of such systems has a substantial impact on their functioning and dynamics[3]. This is the main reason why the quantitative characterisation of complex spatial patterns has received much attention in different fields, from urban studies to biology, from geography to economics, and from transportation to engineering[4–6].

A particularly compelling problem in this field is the quantification of spatial segregation, i.e., the tendency of the units of a system to form uniform agglomerates around closely-located areas (regions, neighbourhoods, census tracts, etc.). The typical example is that of segregation of urban areas by socio-economic indicators, including ethnicity, income, education or religion, which is known to be associated with urban wealth, security, and livability[2,3]. The standard approach in this case is to devise measures of how the local density and heterogeneity of the property under study, as obtained from census data at a given scale, compares with the distribution at the system level, under the assumption that in a non-segregated system the local distribution of, say, ethnicity would closely mirror the overall distribution at the city level[7–13].

There is general agreement about the fact that spatial segregation is a multifaceted characteristic of a system, and that it is quite hard to capture by using a single measure. Indeed, the literature distinguishes different dimensions of the phenomenon[9,14], namely spatial exposure/isolation—the extent to which the members of one group are in close contact with members of another group due to their placement in space—and spatial evenness/clustering—how uniformly groups are distributed in space. Despite this framework being extensively used when referring to spatial segregation patterns, there is no consensus on how these aspects of spatial segregation should be quantified, or on how to compare the levels of segregation in different urban systems[1–3]. In fact, quantifying spatial segregation is still problematic, mainly because most of the measures proposed in the literature depend on the scale at which neighbourhoods are defined, on the granularity of the census data available, or on the presence of free parameters[15,16]. Recently, the hypothesis that the processes behind segregation might be operating at multiple scales simultaneously has gained more support[17–23], though, the question of whether there is a specific spatial scale at which segregation should be measured still remains open. A growing body of literature has recently started examining urban segregation within the paradigm of network science[2,24,25]. This approach consists in analysing one or more graphs associated to an urban

[1]School of Mathematical Sciences, Queen Mary University of London, London, UK. [2]Networks, Data, and Society (NERDS) Research Group, IT University of Copenhagen, Copenhagen, Denmark. ✉e-mail: ssou@itu.dk; v.nicosia@qmul.ac.uk

system—e.g., census tract adjacency, urban transportation, commuting, etc.—and deriving descriptive statistics from the ratios of within-group and between-group connectivity in those graphs[26–30].

Here we propose a principled framework to quantify the multi-level heterogeneity and segregation of a spatial system, and to compare the segregation of different systems, based on the statistics of random walks on graphs[31–35]. We consider the symbolic time series of node properties generated by the trajectories of an unbiased random walker through the graph, and we analyse the spatial distribution of the Class Coverage Time (CCT), that is the expected number of steps required by a random walk to visit a certain fraction of all the classes present in the system, when starting from a generic node. This process is reminiscent with the concept of "zoom lenses" introduced in refs. 21, 23. However, here we do not use any procedure to aggregate population distributions or to compare local abundances with global levels. Indeed, the CCT is purely the result of diffusion on the graph and depends at the same time on the structural characteristic of the graph and on the actual distribution of node properties. A similar approach has been used for instance in ref. 35 to measure exposure to specific classes. We purposely avoid the challenge of directly defining segregation, assuming that the absence of segregation is indicated by the concordance of the statistics of class coverage time with those observed in an appropriate null-model. Moreover, since the measures we propose effectively associate a local level of segregation to each node, they make it possible to compare the local segregation of different areas of the same urban system, or of different urban systems, on equal grounds.

We start by showing how this framework applies to synthetically generated colour distributions on two-dimensional lattices, and then we use it to quantify the ethnic segregation of urban systems in the US and the UK. We find that the distribution of class coverage times provides quite useful insight on the microscopic, meso-scopic, and macro-scopic organisation of ethnicities throughout a city. Furthermore, we find that class coverage times correlate with many deprivation indices, and more strongly than other classical segregation measures do. These results suggest that measuring multi-scale urban segregation by means of random walk statistics is potentially more informative than many of the other current approaches.

## Results

### Model

Let us consider a spatial graph $G(V, E)$ consisting of $N = |V|$ nodes and $K = |E|$ edges[24], and assume that each node $i$ is associated to a certain variable of interest $x_i$, which can in principle be either a scalar or vectorial value. For instance, if nodes represent the neighbourhoods of an urban area, $x_i$ could be the average income of people living in the area represented by node $i$ or the ethnic distribution at node $i$. We are interested in characterising the spatial distribution of $x_i$, that is, to which extent nodes being close to each other in the graph also have similar values of $x_i$ and tend to form homogeneous clusters. In the specific case of urban segregation, we actually want to quantify how homogeneous is the distribution of $\Gamma$ distinct groups across a city, where the groups can represent ethnicities, income classes, education levels, etc. Hence, the variable of interest at each node $i$ is the vector $x_i = \{m_{i,1}, m_{i,2}, ..., m_{i,\Gamma}\}$, where $m_{i,\alpha}$ is equal to the number of citizens of class $\alpha$ living in the census tract associated to node $i$.

Moving from the observation that uniform discrete random walks on a graph preserve a lot of information about the structure of the graph[36–38], we propose to quantify the heterogeneity of the spatial distribution of $x_i$ by means of the temporal statistics of the symbolic dynamics $\{\varphi_{i_0}, \varphi_{i_1}, \varphi_{i_2}, ...\}$ associated to the generic trajectory $\{i_0, i_1, i_2, ...\}$ of a uniform random walk on $G$[34], where $\varphi_{i_t}$ is an appropriately-chosen function of $x_{i_t}$. It is worth stressing that in general $\varphi_{i_t}$ can be constructed in many different ways, according to the specific aspect of segregation that one wants to focus on. In particular, $\varphi_{i_t}$ could depend

not only on the specific quantity $x_i$, but also on the actual number of steps $t$ between the moment the walker started from node $i_0$ and its visit of node $i$.

To illustrate this idea, we assume that we want to quantify the heterogeneity of the distribution of ethnicities across a urban area. Here, each census tract is a node of $G$, and two tracts are connected with a link if they border each other. Each node is associated to a vectorial variable $x_i = \{m_{i,1}, m_{i,2}, ..., m_{i,\Gamma}\}$, representing the distribution of $\Gamma$ ethnicities of citizens living in that tract. To simplify the example, we associate each node to a representative class, corresponding to the most abundant ethnicity in the census tract, so that we can label each tract with one of a finite number of colours. In other words, the variable $\varphi_i$ associated to node $i$ is the most abundant ethnicity found in the corresponding tract. In Fig. 1 we show two possible fictitious distributions of classes superimposed on the map of wards in London. Figure 1a is what a uniformly random distribution of the most abundant ethnicities would look like. In this case, the system is homogeneous, there is no segregation of ethnicities in specific clusters, and the probability that the neighbours of a certain tract belong to any of the available classes does not depend on their position in the map. Conversely, in Fig. 1b we show an artificially imposed clustering of most abundant ethnicities around neighbouring areas. This arrangement of classes is visually more similar to the actual spatial organisation of ethnicities (and of other socio-economic indicators) observed in many modern metropolitan areas, where the emergence of homogeneous clusters is the norm rather than the exception. This second example is the typical spatial pattern that we would consider segregated. Again, remember that the actual association of colours to London wards in the two panels of Fig. 1 is purely illustrative.

Note that a random walker starting at any of the areas in Fig. 1a and moving on the graph $G$ of census tracts will in general require a small amount of time to visit an area characterised by any specific majority ethnicity. For instance, if a walker starts from a red tract, it will normally encounter a light-yellow tract after a small number of steps. This is because the spatial pattern in Fig. 1a is not segregated, as there are no homogeneous clusters of any colour. In other words, a light-yellow tract is available within a small distance of any other tract in the map. Conversely, if a walker starts from one of the tracts of the large red cluster on the right-hand side of Fig. 1b, it will in general require a considerably larger amount of time to visit a light-yellow tract. This is because all the tracts in Fig. 1b are organised in conspicuous spatial clusters. In particular, all the light-yellow clusters are located much

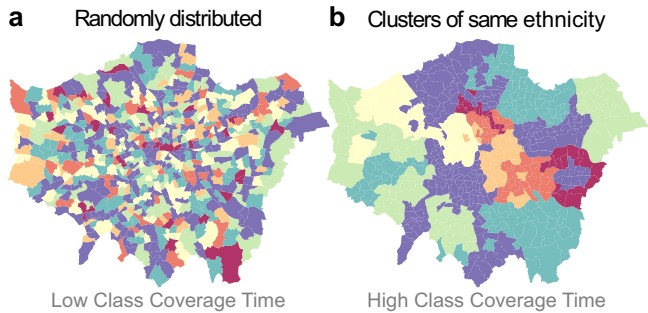

**a** Randomly distributed    **b** Clusters of same ethnicity

Low Class Coverage Time    High Class Coverage Time

**Fig. 1 | Fictitious maps of the association of seven ethnicities to the wards of Greater London. a** The ethnicities are distributed uniformly at random across the city, to simulate a "maximally" homogeneous and unsegregated pattern. In this case, a random walker starting from any ward will get in touch with all the available ethnicities within a relatively small number of steps. **b** The same map with a substantial clustering of ethnicities imposed artificially. In this case, a walker starting in the middle of a cluster will need a lot more time to visit all the other ethnicities. This observation leads to the idea of using the statistics of Class Coverage Time to quantify the level of segregation and heterogeneity of an urban area with respect to a given variable of interest.

farther away from the cluster of red tracts on the right-hand side of the map. This simple example suggests that the number of steps (time) needed to a random walker to leave a homogeneous cluster of nodes, and visit all the other ethnicities represented in a urban area, actually contains very useful information about the spatial organisation of ethnicities across the system.

We propose here to quantify the level of segregation of an urban area with respect to a categorical variable by means of the Class Coverage Time (CCT) of a random walk on the corresponding graph $G$. This is the expected number of steps needed by a walker started at a generic node $i_0$ to visit a prescribed fraction $c$ of all the $\Gamma$ classes present in the system. If classes are distributed evenly across the city, the class coverage time will not depend heavily on the starting node $i_0$. Conversely, if classes tend to form segregated homogeneous groups and clusters, then the amount of steps needed to visit a given fraction $c$ of all the ethnic groups present in the city will actually depend on the starting node, as well as on the shape, size, and depth of the cluster to which the starting node belongs and of the other clusters present in the system. In general, higher heterogeneity in the spatial distribution of class coverage times will correspond to higher and more rigid spatial constraints, and will signal the presence of segregation.

More formally, let us consider a random walk that starts from a node $i$ of $G$ and visits the sequence of nodes $\{i = i_0, i_1, i_2, \ldots, i_t, \ldots\}$ at subsequent discrete time steps $t = 0, 1, 2, \ldots, t, \ldots$. We call $\mathcal{W}_i(t)$ the fraction of distinct classes encountered by the walker up to time $t$ when it started from node $i$ at time 0, and we compute the average over $R$ independent realisations of the walk:

$$\overline{\mathcal{W}_i}(t) = \frac{1}{R} \sum_1^R \mathcal{W}_i(t) \qquad (1)$$

We define the Class Coverage Time (*CCT*) of node $i$ at level $c$ as the expected number of steps after which a walker started at $i$ has encountered a fraction $c$ of the $\Gamma$ classes for the first time, that is:

$$C_i(c) = \arg\min_t \{\overline{\mathcal{W}_i}(t) \geq c\} \qquad (2)$$

We characterise the distribution of Class Coverage Time of a given system by looking at its mean:

$$\mu(c) = \frac{1}{N} \sum_{i=1}^N C_i(c), \qquad (3)$$

its coefficient of variation:

$$\sigma(c) = \frac{\sqrt{\mathrm{Var}(C_i(c))}}{\mu(c)}, \qquad (4)$$

and at the level of local spatial diversity, as measured by:

$$\varrho(c) = \frac{1}{K} \sum_{i=1}^N \sum_{j<i} a_{ij} |C_i(c) - C_j(c)|. \qquad (5)$$

where $\{a_{ij}\}$ are the entries of the adjacency matrix of the graph $G$.

In general, larger values of $\mu(c)$ indicate a more heterogeneous distribution of classes through the system. Similarly, larger values of $\sigma(c)$ correspond to a larger dependence of CCT on the starting point, i.e., $\sigma(c)$ is measuring the overall spatial variance of class coverage times. Finally, larger values of $\varrho(c)$ indicate that neighbouring nodes have very different class coverage times, signalling the presence of local spatial diversity.

As we will show in the following, all these three measures are somehow affected by the relative abundance of each class and by the size of the graph. For this reason, we will consider the average deviation of each quantity from the corresponding quantity measured in a null-model:

$$\Delta\mu = \int_0^1 dc\, |\mu(c) - \mu(c)^{\mathrm{null}}|, \qquad (6)$$

$$\Delta\sigma = \int_0^1 dc\, |\sigma(c) - \sigma(c)^{\mathrm{null}}| \qquad (7)$$

and

$$\Delta\varrho = \int_0^1 dc\, |\varrho(c) - \varrho(c)^{\mathrm{null}}| \qquad (8)$$

The null-model consists of the same graph $G$ as the original system, where the class distributions $\{x_i\}$ have been reassigned to nodes uniformly at random. By doing so, we preserve the overall relative abundance of classes, as well as the way classes tend to be distributed in a single area, but we destroy any existing spatial organisation of classes[2,23] (see Methods for details). The average over the ensemble of random spatial permutations takes into account the many possible ways in which a city can be spatially organised, and removes any bias due to the relative abundance of different classes and any assumption about how the unsegregated spatial distribution should look like. The deviations of $\mu(c)$, $\sigma(c)$ and $\varrho(c)$ from the null-model expectations are a set of principled measures which allow us to compare spatial systems with different number of classes, and characterised by distinct shapes, sizes, and scales. In the following, we call $\Delta\mu$ "spatial heterogeneity", $\Delta\sigma$ "spatial variance" and $\Delta\varrho$ "spatial diversity", for obvious reasons.

**Simple geometries and synthetic class distributions.** In the following sections we explore the behaviour of the three measures based on Class Coverage Time that we have introduced, by considering planar lattices with meaningful pre-assigned class distributions.

**Random class assignments.** We start from the simple case of two-dimensional square lattices—with or without periodic boundary conditions—where each node is associated to one of the $\Gamma$ available classes with uniform probability. In Fig. 2a we report the plot of $\mu(c)$ as a function of the fraction $c$ of classes reached by the walker on $8 \times 8$ (left panel) and $16 \times 16$ grid lattices (right panel) with coordination number equal to 4, for $\Gamma = \{2, 4, 8, 16, 32\}$. As expected, $\mu(c)$ is a non-linear increasing function of $c$, meaning that reaching a higher fraction of the classes becomes harder and harder as $c$ increases. Moreover, $\mu(c)$ is also an increasing function of $\Gamma$ for a fixed fraction $c$, meaning that configurations with more classes typically exhibit larger coverage times, as expected. By comparing the two panels it becomes clear that covering a given percentage $c$ of classes requires comparatively more time on a larger lattice.

These results are quite intuitive and not much surprising, if we consider that even when classes are distributed uniformly at random, local clusters of nodes of the same class eventually emerge. In particular, smaller values of $\Gamma$ and larger lattice sizes have a higher probability of producing larger clusters, which effectively contribute to reducing the probability that the random walk finds a new class at each time step. Indeed, a walker that enters a homogeneous cluster will keep visiting nodes of the same class with high probability, so that it will need more time to find a node belonging to a different class. These observations are confirmed in Fig. 2b, where we report the distributions of cluster sizes for different values of $\Gamma$ (the case $\Gamma = 2$ is in the insets). The larger values of $\mu(c)$ observed in $16 \times 16$ graph in Fig. 2a can indeed be associated to the presence of somehow bigger uniform clusters of the same colour.

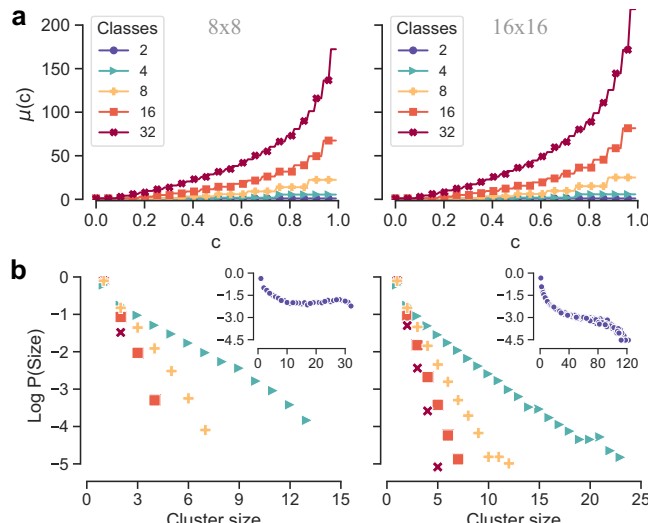

**Fig. 2 | Effect of graph size and number of classes on class coverage times in synthetic spatial networks with Γ classes and random node-class association.**
**a** Mean class coverage time $\mu(c)$ as a function of the fraction of visited classes $c$ on a torus with 64 (left) and 256 (right) cells. Coverage times over 32 classes are significantly larger than those over 2 classes. **b** Size distribution of uniform clusters formed by adjacent nodes of the same class. Larger graphs allow for bigger clusters to emerge, for any number of classes Γ. The legend in **a** marks the corresponding distributions in **b**. The distributions of cluster sizes for Γ = 2 are reported in the insets.

**Large synthetic clusters.** To study the effect of the relative shape and placement of different clusters, we considered finite lattices with pre-assigned class distributions organised in specific patterns. In Fig. 3a we show four arrangements of five classes in a 16 × 16 square lattice without periodic boundary conditions. In each arrangement, four of the classes contain $N/4 − 4$ nodes, and form homogeneous clusters which occupy a quadrant each. The sixteen nodes in the fifth class, instead, (i) either form a single cluster in the centre (top panel), or (ii) in one corner of the lattice (second panel from the top), or (iii) are scattered within another cluster (third panel from the top), or (iv) placed at the four corners (bottom panel). Notice that all these four patterns are associated to the same null-model, since the relative abundances of the five classes are kept constant. However, these four arrangements represent typical stylised distributions, which are the building blocks of more complicated spatial patterns. We report in Fig. 3b–d the values of $\Delta\mu$, $\Delta\sigma$ and $\Delta Q$, respectively, and we show in Fig. 3e, f how these patterns are located in the $\Delta\mu − \Delta\sigma$ and in the $\Delta\sigma − \Delta Q$ planes. Indeed, pattern (i) and (ii) exhibit the largest values of local spatial diversity and spatial heterogeneity (e) and the smallest values of spatial variance (f). This is consistent with the fact that the presence of big clusters is responsible for larger average class coverage times, and makes the CCT of each node depend quite heavily on the actual position of the node inside a cluster, with nodes placed close to the periphery of a cluster normally characterised by smaller values of CCT.

On the contrary, pattern (iii) and pattern (iv) have smaller values of local spatial diversity and larger values of spatial variance (f). This is mainly due to the fact that the nodes in the smallest cluster has fewer internal links to nodes of the same colour. However, the minority class can be found in a smaller number of steps compared to (i) and (ii), as its nodes are spread more uniformly across the domain. Despite patterns (i) and (iv) have a proportion of links to the minority class that is comparable to pattern (ii) and (iii), the clustered pattern in the former yields higher coverage times, as properly detected by $\Delta Q$ and $\Delta\mu$ (e). Indeed, a spatial pattern consisting of an isolated cluster yields the longest trajectories to find other classes, as it was

also noted in ref. 23. Conversely, a more uniform distribution of the minority colour throughout the domain produces a sensible reduction of coverage times.

Note that the profile of $\mu(c)$ for the null-model (dashed black line in Fig. 3b) behaves as expected, i.e., the mean coverage times increase steadily as a function of $c$ up to 0.8 (four classes), mainly due to the uniform abundance and displacement of the four large clusters. We notice a sharp increase for $c \approx 1$, which accounts for the time needed to find nodes belonging to the fifth class, i.e., the smaller one. We provide additional distributions and heat-maps in Supplementary Fig. 1 where the values for $c = 1$ are removed to isolate this effect.

**Effect of domain shape and cluster size.** In Fig. 4a we considered four different random tilings of the same 16 × 16 square lattice considered in Fig. 3, with the aim of isolating the role of the size and shape of local clusters. The tiling consist of 32 classes, respectively organised (from top to bottom) in (i) 32 rectangular clusters of size 2 × 4 (cluster-8), (ii) 64 square clusters of size 2 × 2 (cluster-4), (iii) 64 rectangular clusters of size 1 × 4 (stripe-4) and (iv) 128 rectangular clusters of size 1 × 2 (stripe-2). Notice that configuration (i) (cluster-8) corresponds to the largest possible value of spatial diversity and spatial heterogeneity. On the other hand, configuration (iv) (stripe-2), which is indeed the most similar to the null-model (d-f), yields the smallest values of spatial diversity and spatial heterogeneity (b), as expected. The relative positions of intermediate configurations (ii) and (iii) in the $\Delta\sigma/\Delta Q$ plane (c) can be explained by the fact that a tiling with square clusters provides comparatively lower values of spatial diversity and spatial variance than clusters of four nodes arranged in a line, mainly because a square of size $N$ has a smaller perimeter than a rectangle of the same size, hence more neighbours pointing to clusters of different colour. However, on average the walker needs slightly more time to leave a cluster (ii) than a stripe (iii) of same size (See the relative positions of the two configurations in the $\Delta\mu/\Delta Q$ plane (b)).

Finally, we show that $\Delta\mu$ and $\Delta Q$ are also able to capture differences in the shape of the domain, by considering the 2D lattice with a lateral appendix in Fig. 4g. Indeed, for the same number of classes and the same cluster shapes and sizes as in Fig. 4b, the arrangements in Fig. 4h correspond to much larger values of spatial heterogeneity and local spatial diversity. This is due to two concurring effects. On the one hand, a walker started at the nodes belonging to the lateral appendix will require a much larger amount of time to visit a certain fraction of the classes than the walkers started at nodes in the bulk (see Supplementary Figure 2 for additional details). This is due to the fact that there are very few ways to exit from the appendix and join the bulk. On the other hand, walkers started from nodes in the bulk will have a hard time finding any rare class which is only present in the appendix, since there are very few ways of getting into the appendix from the bulk. The fact that these measures can detect such differences is a quite interesting property, as in metropolitan areas the presence of natural and human-made physical barriers (e.g., rivers, hills, or canals) tends to increase the fragmentation of the social network structure, and produces higher measurable levels of social segregation, as also suggested by a recent work[39].

We have also performed a more in-depth analysis of how well the three measures capture the differences in the typical scale of homogeneous clusters when the relative spatial arrangement of the classes remains the same. In particular, we have considered checkboard-like arrangements of four classes on a 2D lattice, with homogeneous clusters of size 1 × 1, 2 × 2, 4 × 4 and 8 × 8. Also in that case, the different arrangements are associated to the very same null-model, as the relative abundance of colours is preserved across all the configurations. These results are reported in Supplementary Fig. 3, and confirm that all the three measures are sensible to the typical scale of clusters, even when the relative arrangement of colours is kept intact across scales.

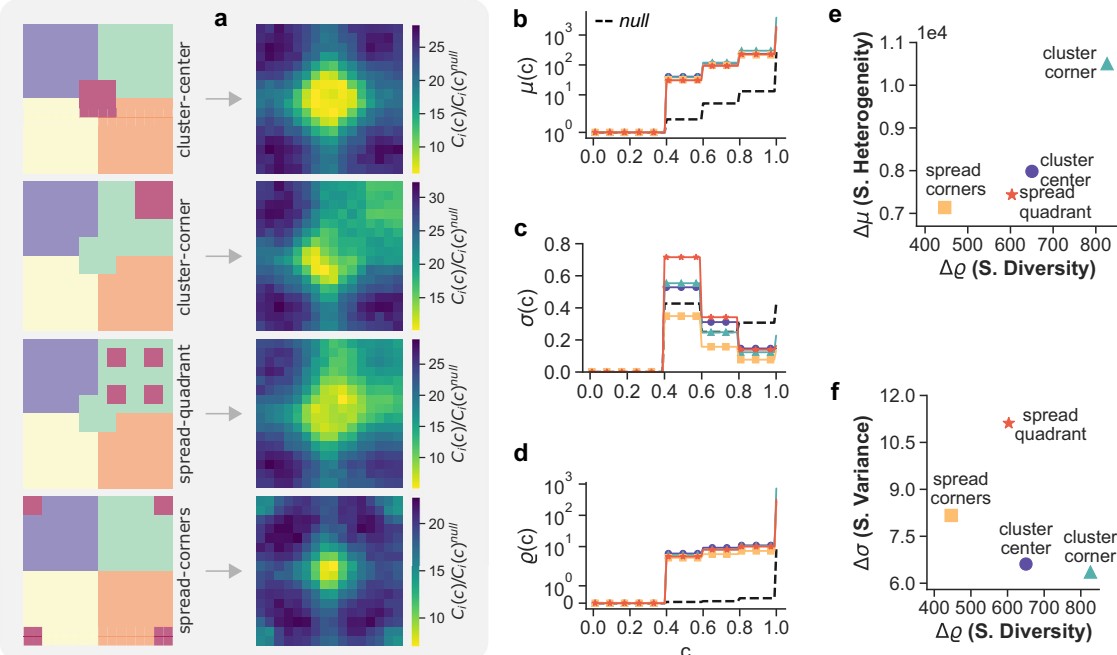

**Fig. 3 | Dependence of class coverage times on size, type and location of homogeneous clusters in a 2-dimensional lattice with synthetic node-class associations. a** The nodes are divided in 4 homogeneous clusters of 60 cells each, placed in the four quadrants, while the remaining 16 nodes in the fifth class are arranged, from top to bottom, as: a central cluster (cluster-center), a cluster in a corner (cluster-corner), spread in one of the quadrants (spread-quadrant), and as four small 4 × 4 clusters on each of the corners of the lattice (spread-corners). The heat-maps reporting the normalised coverage time $C_i(c)/C_i(c)^{null}$ for $c = 0.7$ clearly show the dependency on the starting node. We also report the corresponding profiles of $\mu(c)$ (**b**), $\sigma(c)$ (**c**) and $\varrho(c)$ (**d**), and their values in the corresponding null-model (black dashed lines). The distinct spatial constraints are consistently discriminated by the measure of spatial diversity $\Delta\varrho$ whilst the planes with $\Delta\mu$ **e** and $\Delta\sigma$ (**f**) provide a fuller classification of the patterns.

Despite it is easier and somehow convenient to assign a single measure of segregation to a spatial pattern, we argue that the combination of $\Delta\mu/\Delta\varrho$ and $\Delta\sigma/\Delta\varrho$ provides a more comprehensive and complete picture of the local and global organisation of classes across a given spatial domain, in particular due to the inherent complexity and ambiguity of some spatial patterns of segregation as seen in Figs. 3 and 4.

**Ethnic residential segregation in the US and UK.** After having shown how spatial diversity ($\Delta\varrho$) combined with spatial heterogeneity ($\Delta\mu$) and spatial variance ($\Delta\sigma$) can distinguish different stylised spatial distributions, we show that these measures can be effectively used to quantify and compare the ethnic residential segregation of metropolitan areas. We used geo−referenced census data for metropolitan areas in the US and the UK, and for each urban area we constructed the graph $G$ of physical adjacency between census tracts (US) and wards (UK), respectively. Each node is associated to the distribution of ethnicities in the corresponding area. In the UK data set, ethnicities are divided in 250 classes, while the US data reports 64 different classes (see Methods for details). We computed the coverage time from each node as in Eq. (2), and the corresponding values of spatial variance $\Delta\sigma$, spatial heterogeneity $\Delta\mu$ and local spatial diversity $\Delta\varrho$. In Fig. 5a we report each urban area in the $\Delta\mu/\Delta\varrho$ and $\Delta\sigma/\Delta\varrho$ planes, and a selection of maps showing the values of normalised CCT $C_i(c)/C_i(c)^{null}$, for $c = 0.7$ for each census tract of some representative metropolitan areas (see Methods for more details about the data sets, as well as Supplementary Table 2 and Supplementary Figs. 4–7 for additional information on values for each urban system and their corresponding CCT distributions).

From the analysis of synthetic class distributions, we have learned that higher levels of spatial variance $\Delta\sigma \gg 0$ are associated to more unbalanced spatial distributions of classes. In the case of cities and

ethnicities, this means that citizens experience large variations in the time needed to encounter all the other ethnicities depending on where they live. Conversely, low spatial variance indicates that on average the spatial distribution of ethnicities is relatively uniform across the city and individuals living in different areas are similarly exposed to all the ethnicities present in the system. Low levels of local spatial diversity $\Delta\varrho$ indicate that there is no significant difference on the coverage time of neighbouring areas, that is, the constraints driven by spatial shape are not too important. When $\Delta\varrho \gg 0$, the differences between neighbouring nodes is substantial and segregation is influenced by clusters with similar ethnicity distributions, which indicates the presence of a preference mechanism, often resulting from social or economic pressure. The spatial heterogeneity $\Delta\mu$ is linked directly to the size of the clusters with similar ethnicity composition. In general, smaller clusters yield shorter coverage times, since walkers need less steps to leave them and find other ethnicity compositions in the neighbouring nodes. On the contrary, walkers need more steps to leave larger clusters and find new ethnicities.

We start by noting that Boston is placed at the very far ends of both the $\Delta\mu/\Delta\varrho$ and $\Delta\sigma/\Delta\varrho$ plane, as shown in Fig. 5a. Indeed, the spatial distribution of $\widetilde{C}_i(c) = C_i(c)/C_i^{null}(c)$ across Boston, (map visible in Fig. 5c) shows that the city exhibits clearly opposing patterns of $\widetilde{C}_i(c)$. The blue and green areas, corresponding to regions having smaller values of $\widetilde{C}_i(c)$, i.e., easier access to all the ethnicities, are placed in the southern part of the city, while most of the northern side is characterised by a large number of areas with $\widetilde{C}_i(c)$ up to three times larger than in the null model, indicating the presence of a somehow higher ethnic segregation. Los Angeles, which has a similar wide hot spot of segregated areas in the centre, shows comparably high spatial variance but lower levels of spatial heterogeneity.

Another interesting example is that of Manchester and Sheffield, which have a similar range of $\widetilde{C}_i(c)$ values and are placed closely in the

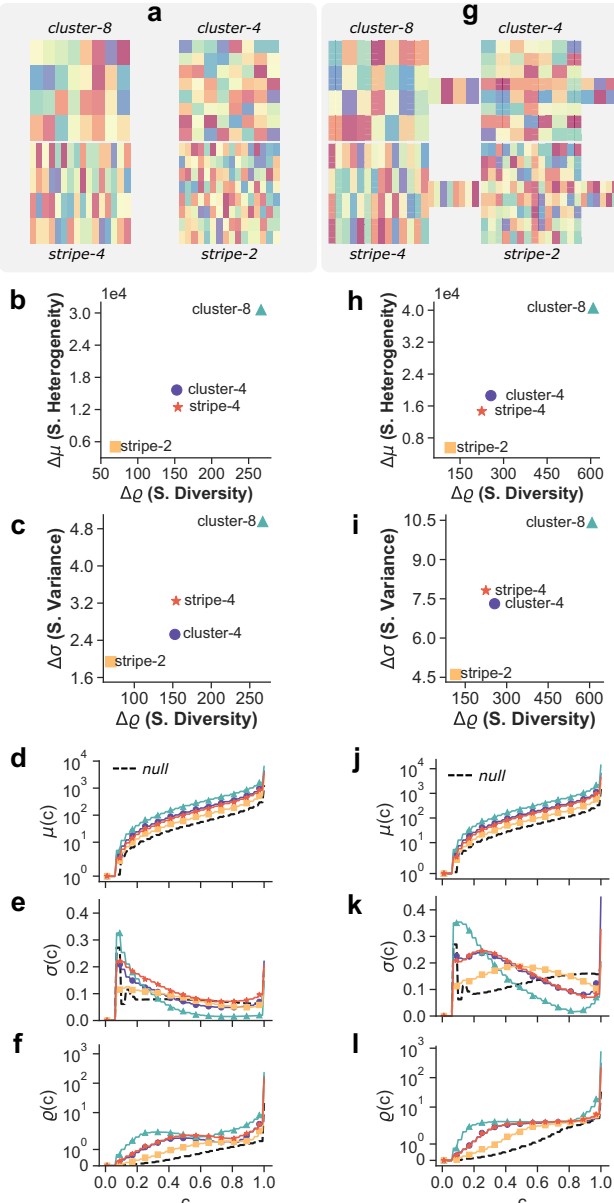

**Fig. 4 | Dependence of class coverage times on domain shape and size and shape of homogeneous clusters. a** The nodes are divided in 32 classes and associated uniformly at random (from top to bottom) to uniform clusters of 8 or 4 cells and to stripes of 4 or 2 cells, on a $16 \times 16$ lattice. **g** Same patterns as in **a**, but on a lattice with a lateral appendix. We also report the corresponding profiles of $\mu(c)$ **d**, **j**, $\sigma(c)$ **e**, **k** and $\varrho(c)$ **f**, **l** and their values in the associated null-model (black dashed lines). The distinct spatial constraints are consistently discriminated in the $\Delta\mu/\Delta\varrho$ plane **b**, **h** while $\Delta\sigma\Delta\varrho$ **c**, **i** provides additional information about neighbouring effect.

$\Delta\sigma/\Delta\varrho$ (a) plane. These similarities can also be confirmed in the corresponding maps. Moreover, Atlanta and the UK Northeast Metropolitan area (indicated as Northeast in the figure) are placed closely in the $\Delta\mu/\Delta\varrho$ plane and show a similar range of $\bar{C}_i(c)$ values. However, we can not observe similarities in the spatial distribution, which is mostly due to the fact that Atlanta is characterised by a much lower value of spatial variance. In this respect, Atlanta has a spatial pattern more similar to Dallas, which might be due to similar historical and urban planning artefacts.

In Fig. 5b we look closely at the behaviour of $\mu(c)$, $\sigma(c)$ and $\varrho(c)$ in London, which is well-known for being characterised by strong ethnic segregation[40,41]. Indeed, there are some areas of the city which clearly

exhibit substantially larger values of class coverage time as indicated by high $\Delta\sigma$. However, the local spatial diversity is relatively low, indicating that adjacent regions tend to be organised in small clusters having very similar distribution of ethnicities. Interestingly, New York shows a quite different organisation, with relatively lower spatial variance and higher spatial diversity, suggesting the presence of larger cluster with very distinct local distribution of ethnicities. Indeed, this is confirmed by the relatively higher values of $\Delta\mu$ in New York (Fig. 5a), indicating that the class coverage times from each of the census tract is comparatively larger than in London.

It is worth noting that all the urban areas analysed in this study present some level of spatial variance or spatial diversity, despite the sizes and population of the areas considered span relatively large ranges. These results are definitely related to the actual distribution of ethnicities across the urban areas with respect to the corresponding null-model (see Supplementary Fig. 8). Interestingly, these result do not seem to depend that much on the granularity at which spatial patterns are sampled, as confirmed by a detrended fluctuation analysis (DFA) of the trajectories of random walks on graphs obtained at different spatial resolutions (see Supplementary Table 1 and Supplementary Fig. 9). A key point of the DFA results is that, despite distinct spatial scales yield small fluctuations, the typical geographic distance at which the random walk still detects large-scale correlations among ethnicity distributions remains quite stable and does not depend on the size of the tracts considered. In other words, this approach based on diffusion effectively removes most of the undesired traditional biases caused by differences in size and spatial scales.

**Relation with socio-economic variables.** The quantification of spatial segregation, especially in urban areas, is not just interesting per se. Indeed, spatial segregation is quite often correlated with other socio-economic indicators, including income level, education, socio-economic deprivation, etc. As a consequence, segregation measures are frequently used as proxies of the wealth, livability, and overall quality of a metropolitan area.

Here we show that the explanatory power of the CCT measures introduced in this paper, in terms of correlations with interesting socio-economic indicators, is consistently higher than that of several other widely used segregation indices. We considered a data set of several socio-economic indicators obtained from the US Census American Community Survey 2011 5-Year estimates[42], including employment, commuting, occupation, income, and security (see Methods for details). We used the Pysal library[43] to compute a variety of classical and more recently devised ethnic segregation measures in US cities, and we correlated them with those socio-economic indicators. The segregation measures are computed on the same data set of US census tracts with ethnicities divided in 64 classes, as in the previous section. We could not consider UK cities as the UK census database does not provide equivalent key statistics at the same spatial scale. The results are reported in Table 1, where we compare the two-sided Spearman correlation coefficient and the associated values of $R^2$ of the diffusion-based segregation measures we introduced here and of a variety of other segregation indices, i.e., Moran's I (MI), Spatial Gini (SG), Spatial dissimilarity (SD), Distance decay exposure (DDE), Distance decay isolation (DDI), Perimeter spatial dissimilarity (PSD) and Boundary spatial dissimilarity (BSD). The $\langle\Gamma\rangle$ index denotes the quantities obtained by averaging the segregation of all classes in a city while $e[\Gamma]$ denotes segregation computed over the entropy of the population distribution at the node.

In general, there is good agreement of our Spatial Diversity $\Delta\varrho$ with $\tilde{\sigma}_{\langle\Gamma\rangle}$[35] and Moran's I $M_{\langle\Gamma\rangle}$. The former is a random walk-based segregation measure that quantifies isolation while the latter quantifies local spatial correlation, which is directly related to the local spatial diversity measured by $\Delta\varrho$. However, $\Delta\varrho$ consistently yields higher values of $R^2$ with better confidence intervals for most of the socio-

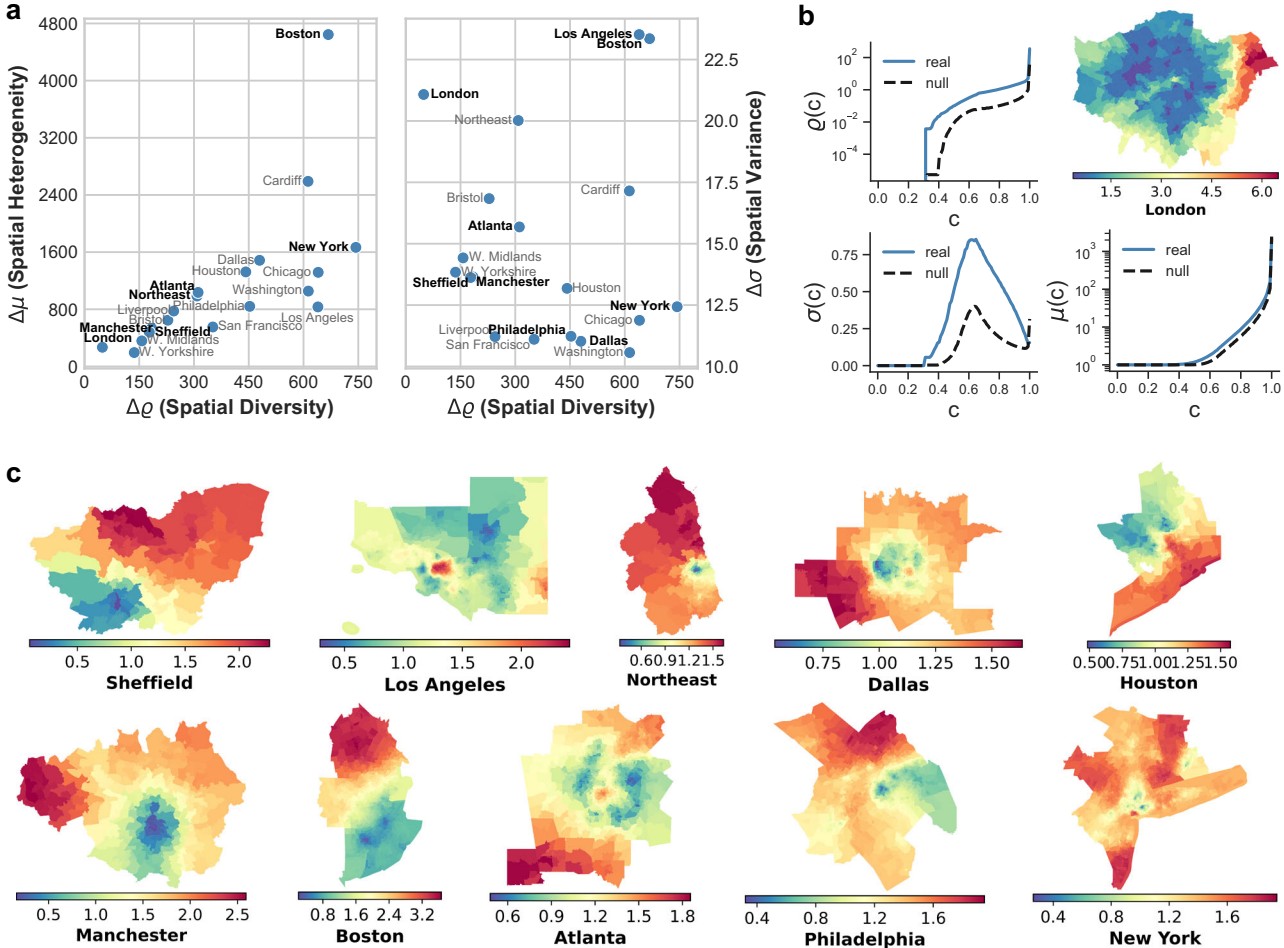

**Fig. 5 | Class coverage times and ethnic segregation in urban systems.**
**a** Metropolitan areas in the US and the UK in the $\Delta\mu/\Delta\varrho$ and $\Delta\sigma/\Delta\varrho$ planes where they are compared by the average deviation from their corresponding null-models. **b** Examples of the class coverage time distributions for London where the spatial diversity $\varrho(c)$, spatial variance $\sigma(c)$, spatial heterogeneity $\mu(c)$ and their values in the corresponding null-model (black dashed lines) are plotted as a function of the fraction of visited classes $c$. **c** Maps of the metropolitan areas marked in bold in **a**. The normalised class coverage time $C_i(c)/C_i(c)^{null}$ for $c = 0.7$ provides detailed insights about the structure of segregation at neighbourhood level.

economic variable considered. In particular, the correlation is remarkably high for Employment, Income, and Social Security.

The high levels of correlations shown in Table 1 suggest with sufficient strength that – in particular agreement with other measures– these socio economic factors surely contribute to the segregation patterns we observe in metropolitan areas. The most relevant result is that the set of measures based on first principles that we have introduced here is able to capture these patterns better than most of the state-of-the-art indicators. By looking at these results, it is not too far-fetched to conclude that diffusion might indeed be capturing some salient aspect behind ethnic segregation that other measures cannot completely encompass. We believe that this is due to the ability of diffusion segregation measures to integrate information at all the relevant spatial scales of a system. However, it is important to stress here that the results shown in Table 1 do not allow us to suggest a causation relation between diffusion-based segregation and measurable socio-economic status, or vice-versa. Such an analysis would require a data set providing consistent longitudinal data about ethnicities and socio-economic indicators, and the formulation of some sort of regressive model to reveal the role of segregation on the variation (increase/decrease) of each indicator. However, understanding the potential impact of diffusion segregation on socio-economic deprivation would definitely be beyond the scope of the present paper, which is methodological in nature.

Hence, we have decided to investigate this interesting relation in a forthcoming work.

## Discussion

Despite we have focused exclusively on the characterisation of ethnic segregation, the methodology introduced here can be used to quantify the spatial variance and spatial diversity of the distribution of any categorical variable, including socio-economic indicators like income, access to services, education level, and so forth[44]. The consistent behaviour of $\Delta\mu$, $\Delta\sigma$ and $\Delta\varrho$ across different scales is indeed a very desirable property of segregation measures, as also pointed out by both classical and more recent works[45,46]. The fact that these measures are appropriately normalised by comparing with the corresponding null-models, make them suitable for comparing the spatial heterogeneity of the same variable in different systems, irrespective of their peculiar size and shape, of the actual number of different classes or categories available in each system, and of the granularity at which spatial information is aggregated.

The framework proposed here is quite flexible and extensible. Although we have mostly focused here on the time to find a certain ethnic group, irrespective of its relative abundance, one can refine the analysis by using an appropriate definition of the node quantity $\varphi_{i,t}$ to take this aspect into account, e.g., by setting it equal to the vector of local abundances of each ethnic group. One possible shortcoming of

**Table 1 | Two-sided Spearman correlations of socio-economic variables for US metropolitan areas with CCT quantities and other segregation measures**

| Soc.-Econ. variables | Diffusion | | | | Moran I | | Sp. Gini | | Distance decay | | Spatial dissimilarity | | |
|---|---|---|---|---|---|---|---|---|---|---|---|---|---|
| | $\Delta\mu$ | $\Delta\sigma$ | $\Delta Q$ | $\bar{\sigma}_{(\Gamma)}$ | $MI_{e[\Gamma]}$ | $MI_{(\Gamma)}$ | $SG_{e[\Gamma]}$ | $SG_{(\Gamma)}$ | $DDE_{(\Gamma)}$ | $DDI_{(\Gamma)}$ | $SD_{(\Gamma)}$ | $PSD_{(\Gamma)}$ | $BSD_{(\Gamma)}$ |
| **Employment** | | | | | | | | | | | | | |
| Employed | 0.02 | 0.00 | 0.71*** | 0.33* | 0.35* | 0.40** | 0.09 | 0.11 | 0.01 | 0.35* | 0.16 | 0.16 | 0.16 |
| Unemployed | 0.01 | 0.05 | 0.44** | 0.27 | 0.30* | 0.73*** | 0.07 | 0.05 | 0.01 | 0.23 | 0.06 | 0.06 | 0.06 |
| **Commuting** | | | | | | | | | | | | | |
| Car/truck/van (alone) | 0.02 | 0.01 | 0.71*** | 0.35* | 0.33* | 0.28 | 0.09 | 0.10 | 0.03 | 0.35* | 0.15 | 0.15 | 0.15 |
| Public transportation | 0.00 | 0.00 | 0.42** | 0.21 | 0.29 | 0.42** | 0.16 | 0.08 | 0.00 | 0.16 | 0.10 | 0.10 | 0.10 |
| Walked | 0.04 | 0.09 | 0.73*** | 0.45** | 0.37* | 0.59*** | 0.16 | 0.16 | 0.03 | 0.52** | 0.17 | 0.17 | 0.17 |
| Other means | 0.00 | 0.02 | 0.32* | 0.05 | 0.33* | 0.40** | 0.01 | 0.01 | 0.09 | 0.33* | 0.00 | 0.00 | 0.00 |
| Worked at home | 0.02 | 0.01 | 0.25 | 0.06 | 0.42** | 0.52** | 0.00 | 0.01 | 0.00 | 0.29 | 0.00 | 0.00 | 0.00 |
| **Occupation** | | | | | | | | | | | | | |
| Management/science/arts | 0.00 | 0.00 | 0.61*** | 0.37* | 0.39* | 0.45** | 0.14 | 0.06 | 0.00 | 0.39* | 0.09 | 0.09 | 0.09 |
| Service | 0.01 | 0.01 | 0.69*** | 0.42** | 0.32* | 0.45** | 0.13 | 0.14 | 0.02 | 0.36* | 0.16 | 0.16 | 0.16 |
| Sales and office | 0.04 | 0.00 | 0.73*** | 0.33* | 0.32* | 0.36* | 0.13 | 0.10 | 0.00 | 0.25 | 0.17 | 0.17 | 0.17 |
| Prod./transp./material | 0.12 | 0.25 | 0.44** | 0.28 | 0.00 | 0.13 | 0.02 | 0.35* | 0.00 | 0.04 | 0.28 | 0.28 | 0.28 |
| **Occupation Industry** | | | | | | | | | | | | | |
| Retail trade | 0.03 | 0.03 | 0.73*** | 0.33* | 0.39* | 0.56*** | 0.14 | 0.10 | 0.00 | 0.30* | 0.16 | 0.16 | 0.16 |
| Finan./real estate/leasing | 0.07 | 0.03 | 0.79*** | 0.36* | 0.33* | 0.45** | 0.16 | 0.13 | 0.00 | 0.27 | 0.22 | 0.22 | 0.22 |
| Profe./scient./manag./adm. | 0.00 | 0.00 | 0.40** | 0.19 | 0.39* | 0.42** | 0.01 | 0.00 | 0.00 | 0.39* | 0.01 | 0.01 | 0.01 |
| Edu./health/social care | 0.05 | 0.06 | 0.82*** | 0.61*** | 0.30* | 0.56*** | 0.24 | 0.21 | 0.00 | 0.45** | 0.25 | 0.25 | 0.25 |
| Arts/entert./accom./food | 0.00 | 0.00 | 0.54** | 0.24 | 0.33* | 0.44** | 0.05 | 0.04 | 0.00 | 0.25 | 0.07 | 0.07 | 0.07 |
| **Income (USD)** | | | | | | | | | | | | | |
| <$10,000 | 0.10 | 0.12 | 0.77*** | 0.69*** | 0.08 | 0.32* | 0.37* | 0.39* | 0.03 | 0.15 | 0.42** | 0.42** | 0.42** |
| $10,000-$14,999 | 0.02 | 0.19 | 0.50** | 0.50** | 0.03 | 0.59*** | 0.08 | 0.35* | 0.00 | 0.11 | 0.29 | 0.29 | 0.29 |
| $15,000-$24,999 | 0.14 | 0.20 | 0.59*** | 0.45** | 0.01 | 0.33* | 0.07 | 0.45** | 0.00 | 0.07 | 0.40** | 0.40** | 0.40** |
| $25,000-$34,999 | 0.20 | 0.10 | 0.67*** | 0.50** | 0.00 | 0.14 | 0.10 | 0.50** | 0.01 | 0.06 | 0.47** | 0.47** | 0.47** |
| $50,000-$74,999 | 0.05 | 0.01 | 0.77*** | 0.45** | 0.24 | 0.32* | 0.20 | 0.16 | 0.01 | 0.23 | 0.24 | 0.24 | 0.24 |
| $75,000-$99,999 | 0.00 | 0.01 | 0.65*** | 0.39* | 0.35* | 0.50** | 0.16 | 0.10 | 0.00 | 0.30* | 0.14 | 0.14 | 0.14 |
| $100,000-$149,999 | 0.01 | 0.00 | 0.65*** | 0.40** | 0.36* | 0.40** | 0.11 | 0.09 | 0.01 | 0.45** | 0.11 | 0.11 | 0.11 |
| >$200,000 | 0.01 | 0.00 | 0.42** | 0.28 | 0.33* | 0.42** | 0.02 | 0.02 | 0.03 | 0.47** | 0.02 | 0.02 | 0.02 |
| **Supplementary Security** | | | | | | | | | | | | | |
| Social Security | 0.01 | 0.06 | 0.71*** | 0.65*** | 0.19 | 0.59*** | 0.28 | 0.24 | 0.00 | 0.24 | 0.28 | 0.28 | 0.28 |
| Cash public assist. | 0.00 | 0.04 | 0.47** | 0.73*** | 0.06 | 0.59*** | 0.17 | 0.23 | 0.00 | 0.19 | 0.20 | 0.20 | 0.20 |
| Food Stamp/SNAP 12m | 0.12 | 0.13 | 0.75*** | 0.67*** | 0.04 | 0.30* | 0.32* | 0.42** | 0.04 | 0.10 | 0.45** | 0.45** | 0.45** |

The spatial heterogeneity, spatial variance, and spatial diversity of a city show a remarkably consistent association with several socio-economic indicators, especially those related to occupation, employment, income, and security. As a result, those indicators, and in particular $\Delta Q$, are good candidates to work as a proxy of socio-economic segregation and of the quality of life of an urban area.

$*p < 0.1$; $**p < 0.05$; $***p < 0.01$. Actual p-values are reported in Supplementary Table 3.

using a very detailed classification of ethnicities is that rare ethnicities will bias the class coverage times towards higher values (see Supplementary Fig. 8). In some extreme situations, a walker will take a considerably large amount of time to encounter a given ethnicity if there are only a handful of citizens belonging to it, and they are all concentrated in one node. Nevertheless, a similar approach has recently been shown to be particularly useful when dealing with coarse-grained ethnic data[44], where the absence of rare ethnicities avoids the emergence of spurious effects on the statistics of the random walk. We believe that the choice we made avoids the many potential biases introduced by aggregating all the rare ethnicities in a small number of arbitrary classes, since these aggregations vary across countries and administrations. Indeed, the presence of a large number of ethnic classes is not a limiting factor when analysing urban spatial segregation through diffusion, while it might introduce serious biases in the computation of other indicators.

It is important to stress here that the scale at which segregation is measured and quantified is a fundamental aspect of the problem. In particular, quantifying the segregation of a specific neighbourhood of a city makes as much sense as assessing the overall segregation level of a much larger geographical region that contains several metropolitan areas. When we say that diffusion-based segregation measures allow to wash out size-dependence, we do not intend to say that those measures are insensible to the differences between a large metropolitan area and one of its neighbourhoods. Instead, we refer to the importance of considering the issues arising when comparing the segregation of two areas characterised by different spatial scales, e.g., the segregation of East London at the level of wards and the segregation of Greater London at the level of boroughs. We acknowledge that no researcher working on segregation would be tempted to compare these two systems, as their relative scales are obviously distinct, and the mechanism responsible for the emergence of segregation are different at the level of neighbourhood and regions.

However, the same problem potentially arises when one compares the segregation of more than one different metropolitan areas, using data coming from more than one single census source. In other words, scale-dependence is at work when we compare the segregation levels of London, UK, with those of Houston, TX, or with Sao Paulo, Brazil. Even considering only the case of UK and US cities, there are substantial differences in the definition of census tracts and in the granularity of ethnicity classification, as well as differences in the total number of distinct ethnicities: the UK census considers as many as 250, while the US census has only 64 classes. By employing a normalisation with respect to a meaningful null-model, the diffusion-based measures of segregation proposed in this work allow to take into account this kind of size-dependence, thus allowing us to make a fair comparison between, say, a city in the UK and a city in the US.

A very interesting finding of this study is that the measures based on Class Coverage Times signal the presence of some level of segregation in all systems, independently of their shape and size. This is in line with some recent results[20] showing that the tendency of high income households to cluster increases with the size of the city, but is present in all the analysed settlements. The fact that none of the urban systems analysed in this study presents a distribution of Class Coverage Times compatible with the corresponding null-model suggests that segregation is a robust emergent phenomena. Even when it is not determined by actual spatial constrains, it is most often driven by the local population distribution, and by some intrinsic aggregation dynamics that let a segregated pattern emerge and consolidate over time. Finally, the high association between the distribution of Class Coverage Times and several socio-economic indicators potentially allows us to use the diffusion properties of a metropolitan area as a principled and reliable proxy for the livability of a city.

## Methods

### Ethnicity data

We used the UK Office for National Statistics 2011 Census quick statistics tables, which include population estimates classified by ethnic group. The available territorial divisions are regions, districts, unitary authorities, MSOAs, LSOAs and OAs in England and Wales. The households are divided in 250 ethnic groups for the detailed tables. All data of the 2011 British Census are available from the Office for National Statistics[47]. The delineations of the statistical areas are available from the UK Data Service[48].

For US cities, we used the American Census Bureau's 2010 Decennial Census data[49], which include race/ethnicity of individuals at the Census Block level. The households are divided in 64 race/ethnic groups for the detailed tables within the corresponding combined statistical area. The delineations of the Census blocks are available from the same agency at the Geography section[50]. For metropolitan areas containing islands as part of the territory, we focused on the largest surface to avoid the presence of disconnected components on the corresponding graph.

**Socio-economic indicators data set.** To compute the correlations with socio-economic variables reported in Table 1 we used the American Community Survey (ACS), 2011 5-Year Estimates Data Profiles, which is integrally connected with the US Census 2010. The 5-Year estimates provide all tabulation areas irrespective of population size. The ACS is a nationwide survey conducted by the Census Bureau designed to provide more frequently updated demographics for national and sub-national geography than provided by the decennial census program. It collects and produces population and housing information at census tract up level every year, and has an annual sample size of about 3.5 million addresses.

**Random walk trajectories and CCT profiles.** The computation of CCT profiles are based on extensive simulations of independent random walks on the adjacency graph of wards/tracts, keeping track of the fraction $\mathcal{W}_i(t)$ of classes visited up to each step $t$ of each walk. We considered $R = 1000$ independent walkers starting from each of the nodes of a graph, and we stop a walk only when it has visited all the classes in the system. All the trajectories starting from the same node are padded to the length $T_i$ of the longest of those trajectories. In practice, if a realisation of the walk reaches all the classes in $\tau < T_i$, then we set $\mathcal{W}_i(t) = 1$ for all $\tau < t \le T_i$ for that trajectory. This simply means that if a walk would continue after having visited all the classes at time $\tau$, the value of $\mathcal{W}_i(t)$ for $t > \tau$ will remain equal to 1. We compute the CCT profile of node $i$ by considering, for each time $t$, the average over the $R$ realisations of $\mathcal{W}_i(t)$.

**Null model.** Given a graph $G$ and an assignment of classes to nodes, we considered the null model where node class distributions are randomly reassigned while preserving the structure of the graph $G$ and the local population distribution at each node. It is worth noting that the spatial scale at which the null-model is defined is the same of the system under study, so that problems generated by comparing cities at different scales are reduced to a minimum.

**Synthetic systems.** The synthetic systems presented in the first example correspond to grids of 256 and 64 cells with periodic boundary conditions, where each node is associated to a single class, chosen uniformly at random. The CCT for each node is averaged over 1000 trajectories, and the results shown are averaged across 100 realisations of class assignments for each setup. Similarly, for each of the synthetic geometries shown in Figs. 4 and 3, the null-model was obtained by considering 100 independent realisations of the corresponding class assignment.

**Spurious effects for c ≃ 1.** Coverage time distributions for a city will in general depend on the abundance of the classes and on how they are distributed in space. In particular, if a class is very rare, i.e., present only in a few tracts, then the time needed to visit all the classes will effectively become comparable with the cover time, which is known to scale exponentially with the size of the graph[34]. We decided to minimise these spurious effects by removing from the analysis of CCT the case $c = 1$.

**Measures of segregation.** The set of measures of spatial segregation reported in Table 1 include:

– The random walk based normalised segregation index[35] denoted by $\tilde{\sigma}_{(\Gamma)}$ is defined as the probability that an individual meets an individual from the same social group.
– The classical Spatial Dissimilarity (SD) index[51] which can be interpreted as a measure of how different the social composition of neighbourhoods is, on average, from the social composition of the study area.
– Two variations of spatial dissimilarity index, the Boundary (BSD) and Perimeter/Area ratio (PSD)[52] which take into account the length of the common boundary between two areal units and their shapes.
– The Distance Decay Exposure (DDE) and the Distance Decay Isolation (DDI) indices proposed by ref. 53. The former indicates the probability that an individual belonging to a group meets anywhere in space someone from other group while the later accounts for the probability of meeting someone from the same group.
– The Spatial Gini (G) index, that infers the contribution of spatial neighbouring pairs to overall inequality across a set of regions. We considered the share of inequality in non-neighbour component to obtain the correlations[54]. The entropy-based version was computed using as node variable the Shannon entropy of the ethnicity distribution in the corresponding area, denoted here by $x_i$. It is given by:

$$SG = \frac{\sum_{i=1}^{N}\sum_{j=1}^{N} w_{ij}|x_i - x_j| + (1 - w_{ij})|x_i - x_j|}{2n^2 \langle x \rangle} \tag{9}$$

where $N$ is the number of neighbourhoods and $\langle x \rangle = \frac{1}{N}\sum_i x_i$ is the mean of the variable of interest. The spatial weight $w_{ij}$ is defined according to the adjacency matrix $A$ where $w_{ij} = 1$ if two areas are neighbours, and 0 otherwise. The diagonal elements $w_{ii} = 0$ as defined in $A$ and $W$ corresponds to the sum of all weights.

– The Moran's I (M) Global Auto-correlation, which measures spatial auto-correlation based on both feature locations and feature values simultaneously[55]. Similarly to Gini, the entropy based values were given by:

$$I = \frac{N}{W} \frac{\sum_i \sum_j w_{ij}(x_i - \langle x \rangle)(x_j - \langle x \rangle)}{\sum_i (x_i - \langle x \rangle)^2} \tag{10}$$

The indices considered row standardisation of the spatial weights matrices which were based on binary associations, i.e., 1 for neighbouring areas and 0 otherwise. The measures were computed using the PySAL package[43]. For all measures (except entropy-based Gini and Moran), we obtained the value for each of the $\Gamma$ classes and computed the average, so that the comparison with the diffusion segregation is meaningful.

**Reporting summary**
Further information on research design is available in the Nature Research Reporting Summary linked to this article.

## Data availability
The ethnicity data that supports the findings of this study section is openly available at the respective Census agencies[47,49]. A cleaned version of the input files is available in the repository: https://github.com/segregation-rw/ethnic-segregation-rw. The data generated by the random walk process for all experiments are available in the repository: https://doi.org/10.5281/zenodo.5521053.

## Code availability
The code to simulate the random walk process on geographic networks and synthetic systems is open source and available in the Github repository: https://github.com/segregation-rw/ethnic-segregation-rw. We have also made a static release in the repository https://doi.org/10.5281/zenodo.6874241

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

## Acknowledgements
We thank Aleix Bassolas, Silvia Rognone, and Andrea Santoro for useful discussions and feedback. VN acknowledges support from the EPSRC New Investigator Award Grant No. EP/S027920/1. SS acknowledges support by a research grant (project number: 00037394) from Villum Fonden. This work made use of the MidPLUS cluster, EPSRC grant no. EP/K000128/1. This research used Queen Mary's Apocrita HPC facility, supported by QMUL Research-IT. doi.org/10.5281/zenodo.438045.

## Author contributions
S.S. and V.N. devised the study. S.S. performed the computations. S.S. and V.N. contributed to methods, analysed the data, and wrote the manuscript.

## Competing interests
The authors declare no competing interests.
