## [Peer Review File · Nature Communications]

Reviewers' Comments:

Reviewer #1:

Remarks to the Author:

The authors study random-walk based measures of spatial segregation. They build on tools developed for network analysis, applying them here to spatial networks.

Consider a network with a given adjacency matrix and assume that each nodes belong to one of Γ classes. Then, the "time" – ie the number of steps – that it will take for a (random or non random) walker starting from node i to visit at least one node from each class will depend on the (spatial) heterogeneity of the network. If the classes are well mixed across the network, then starting from any node one quickly visits every class. On the contrary, if there are strong patterns of concentration, with nodes grouped together according to class, then it will take a much longer time, on average, to cover all classes, especially for walkers starting from nodes located at the heart of 'ghettos' (that is, low-diversity regions of the network).

Although the authors bring a more formal perspective to this idea (making a more thorough use of random-walk based network analysis), it is not new, and related measures have been explored recently in the literature, notably in geography. There, different choices were made as to the features to be retained in order to produce measures of segregation from walk-based exploration of spatial data and I do think that the authors would need to engage more with the existing literature. This would certainly lead to a more detailed analysis of their own choices, and perhaps to some modification of these.

Indeed while it is true that there is a need for what the authors call "a principled framework to quantify the level of segregation", in my opinion the manuscript falls short of really providing such a framework. There are two main reasons for this shortcoming:

1. Some of the author's choices limit the scope of the method and might even appear as missing certain points one wants to address when measuring segregation: for instance, is class coverage time really the most relevant indicator in a random walk exploration of the network? If a city is composed of 25% group A and 75% group B, is the time it takes a random walk started on a group-A node to meet a group-B individual a good indicator, or is it more relevant to look at the time it takes for a random walker to have

a good approximate idea of the proportions of each group on the whole network? (The latter is a measure that was proposed and used in several recent articles – see eg J. Östh, W. Clark & B. Malmberg, *Measuring the scale of segregation using k-nearest neighbor aggregates*, *Geographical Analysis* 47:1, 34-49 (2015); A. Petrović, M. van Ham & D. Manley, *Multiscale Measures of Population: Within- and between-City Variation in Exposure to the Sociospatial Context*, *Annals of the American Association of Geographers*, 108:4, 1057-1074 (2018); A. Petrovic, *Multiscale spatial contexts and neighbourhood effects* (2020); as well as refs. [6] and [30] in the manuscript.) Also, the behaviour of the proposed indices on synthetic configurations is not fully analyzed. For instance on Fig. 3, one should explain the curves obtained for the null model and comment the artefacts that are visible on them. Are these desirable? Do they lead to any bias in the results?

2. The authors do not provide a clear analysis of what their measures capture and what they do not capture. They show that their pair of indices takes different values on visibly distinct synthetic patterns, and then they merely register the fact these indices lead to a classification of UK and US cities that differs from the one obtained using the I-G pair of standard spatial indices (Moran's I and the spatial Gini index). But why is it the case? And, then: which pair of indices gives the "correct" results? It is indeed also the case that the I-G pair distinguishes between visually different spatial configurations. Now, I am acting a bit as the devil's advocate here, as I am

convinced that walk-based measures are better – although not necessarily with the indicators that the authors choose here to retain from a random walk exploration of the network. But in any case, the relevance of the results obtained with new indicators should be ascertained. In my opinion, this would best be done through interdisciplinary work – that would allow to interpret the new results, decide whether they make sense or not and, most importantly, put them in perspective with existing knowledge. There are many more indices than the I-G ones that are in use today in the city science literature (be it in geography, demography, sociology or economics – see e.g. *New Methods for Measuring and Analyzing*

Segregation, M. Fossett, Springer Series on Demographic Methods and Population Analysis (2017)). When adding to the "jungle" of segregation indices, one needs to thoroughly analyze and specify what novel desired or desirable features one's new measure is bringing to segregation analysis. This issue is only very partly addressed in the manuscript.

These two points lead me to recommend major revisions before the manuscript be re-considered for publication.

Reviewer #2:

Remarks to the Author:

Summary:

This study uses the properties of spatial graphs in order to derive a novel measure of (ethnic) segregation. Given a graph $G(V, E)$ in which vertices correspond to individuals (i.e, the ethnic majority in a census tract) and edges represent connections (i.e, neighboring census tracts), the "Class Coverage Time (CCT)" $\mu(c)$ is defined as the expected number of steps required by a random walker to visit a fraction c of ethnicities. With the help of simulations, properties of the measure are established. Finally, census tract data from the US and UK is used to compare the CCT with other prominent measures in the literature (Moran I and Spatial Gini Coefficient).

Comments:

The CCT is a very intuitive segregation measure that fits nicely into the existing literature. The following points may be considered for improvements:

- The study emphasizes the indices of "spatial variance" $\Delta\sigma(c)$ and "spatial diversity" $\Delta\rho(c)$, which both connect to the distribution of the CCT at the node level. For example, Figure 3 shows that both $\sigma(c)$ and $\rho(c)$ are higher in the cluster-cornered than in the cluster-centered case. According to my understanding, any segregation measure should order the different cases as follows: cluster-centered < (spread-quadrant, spread-corners) < cluster-corners. This suggests that the analysis should focus on $\mu(c)$.
- The application about ethnic segregation in the US and UK shows that the representation of the cities in the Moran I/Spatial Gini Coefficient space is different from the representation in the $\Delta\rho/\Delta\sigma$ space. Since the approaches are different, this is probably not too surprising. Given the vast amount of measures that have been proposed, we have reached a point where we have to think beyond a comparison of intuitive measures. What characterizes a reasonable segregation measure? One way to address this question is by means of an axiomatic analysis (normative approach). Another way would be to show that particular measures have empirical relevance, that is, one should interpret ethnic segregation as a personal characteristic and establish that it affects other variables. Finally, one could try to derive the analytic properties of the measure (express the measure as a function of the matrix that represents the spatial networks).
- In terms of comparisons with other measures, the natural benchmarks are the measures proposed by Echenique and Fryer (2007) and Ballester and Vorsatz (2014).

Reviewer #3:

Remarks to the Author:

This is an interesting, nicely written, piece of work. Using well-known network techniques it creates a graph from a spatial distribution of administrative areas for several cities. With the help of a random walker, it proposes another segregation measure that it is comparable with classical segregation measures, like Moran I and spatial Gini.

I particularly enjoy that the mathematical formalism as is clear and easy to reproduce. More importantly, the presented framework can be extended to accommodate other urban economic-social measures. This is for me, the real value of this work. The actual application to measure ethnicity segregation is interesting but it is not clear how it stands among the whole ecosystems of

ethnicity segregation measures. Although the authors address the differences between theirs and the classical measures, is not clear the underlying mechanism behind these differences. I will suggest discussing in more mathematical detail the possible reasons behind these differences values among the three measures. Besides that, I have no objection to recommending this work for publication.

Dr Roberto Murcio

Response to Referee: 1

=====

We would like to thank the reviewer for the comments and remarks. We acknowledge that the results on synthetic system and the analysis of real world systems could be better explained. In the revised manuscript we have addressed all the comments of the referee in a constructive way, as detailed in the point-to-point response below.

> Some of the author's choices limit the scope of the method and might
> even appear as missing certain points one wants to address when
> measuring segregation: for instance, is class coverage time really
> the most relevant indicator in a random walk exploration of the
> network? If a city is composed of 25% group A and 75% group B, is
> the time it takes a random walk started on a group-A node to meet a
> group-B individual a good indicator, or is it more relevant to look
> at the time it takes for a random walker to have a good approximate
> idea of the proportions of each group on the whole network?

The main reason for not taking into account ethnicity abundance is that we used the highest possible accuracy data about ethnicities, which consists of 250 classes in the UK and 64 classes in the US. We believe that this choice present a more objective picture of the ethnic diversity in a urban systems. This becomes evident in Supplementary Fig S7, by comparing the availability of some ethnicities in London to the other UK cities. To our knowledge, no segregation study has considered so many ethnic groups and we believe our approach contribute to this gap in knowledge.

As a result of this choice, most ethnicities are relatively rare, and the computation of CCT will be mostly affected by the rarest ethnicities, e.g., those that are present in low proportion in just one cell. At the same time, we did not want to impose a strict definition of segregation, which is instead implied by the proposal of comparing the time needed to a walker to get an idea of the proportions of each group in the whole network. This would impose a quite strong constraint on the kind of segregation we measure. We also wanted to avoid the many potential biases introduced by aggregating all the rare ethnicities in a small number of arbitrary classes, since these aggregation vary a lot across countries and administrations.

Nevertheless, the proposed framework allows to incorporate any kind of information, by means of an appropriate definition of the variable of interest ϕ_i . We have provided an example in the Supplementary material, where we looked at the Detrended Fluctuation Analysis of the local Shannon entropy of ethnicities at the nodes, showing that the typical (spatial, real) scale of correlations in this measure does not depend on the specific scale at which cells are defined.

We have also considered the discrepancy between local and global ethnicity distributions by means of the Jensen-Shannon divergence on a modified version on our method (Bassolas, A., Sousa, S. & Nicosia, V. J. R. Soc. Interface 18, 20200961 (2021)). This approach has been shown to be quite useful when working with coarse-grained ethnic data, but the difference between the curves is computed with a parametric precision level, and the dependency of the results on this choice still needs to be fully analysed.

> Also, the behaviour of the proposed indices on synthetic
> configurations is not fully analyzed. For instance on Fig. 3, one
> should explain the curves obtained for the null model and comment
> the artefacts that are visible on them. Are these desirable? Do they
> lead to any bias in the results?

We edited that section to better explain the behaviour of the measures. In particular we split the example with 5 classes from the 32 classes case. We included new numerical simulations with increased number of cells assigned to the "minority" class, which indeed shows a clearer effect of shape and location on coverage times. This is

confirmed by the newly included spatial heterogeneity distributions where values increase steadily as a function of the fraction of visited classes c , which is consistent with the coverage time estimates in Fig.2, i.e., the more classes the harder to find them. We show that in fact, the analysis of the three measures combined provide a more complete picture of the spatial complexity of the system.

We improved the discussion to better explain the behaviour of the null model, where mean coverage times increase steadily as a function of c up to 0.8 (4 classes), but soars when the minority class is considered. This is expected, since the effect of the oddly distributed minority class is negligible up to when all the 5 classes are considered. The null model is an average over the ensemble of random spatial permutations of the city, which encompasses several possible ways in which a city can be spatially organised. This minimise any bias or assumptions about how the unsegregated city should look like. We also included a new Fig. (Figure 4 in the revised text) for the synthetic example with 32 classes, where the added spatial heterogeneity distributions clearly shows the effect of the fraction of visited classes c , and how they compare with the null model.

> The authors do not provide a clear analysis of what their measures
> capture and what they do not capture. They show that their pair of
> indices takes different values on visibly distinct synthetic
> patterns, and they then merely register the fact these indices lead
> to a classification of UK and US cities that differs from the one
> obtained using the I-G pair of standard spatial indices (Moran's I
> and the spatial Gini index). But why is it the case? And, then:
> which pair of indices gives the "correct" result? It is also the
> case that the I-G pair distinguishes between visually different
> spatial configurations. And, in any case, the relevance of the
> results obtained with new indicators should be ascertained. In my
> opinion, this would best be done through interdisciplinary work --
> that would allow to interpret the new results, decide whether they
> make sense or not and, most importantly, put them in perspective
> with existing knowledge ... When adding to the "jungle" of
> segregation indices, one needs to thoroughly analyze and specify
> what novel desired or desirable features one's new measure is
> bringing to segregation analysis. This issue is only very partly
> addressed in the manuscript.

We thank the referee for this comment, which indeed inspired us to completely reshape that section and the corresponding discussion. In particular, the revised manuscript includes an in-depth analysis of how the measures we propose and other classical and more recent segregation measures correlate with some socio-economic indicators which are considered connected to spatial and social segregation. We restricted our analysis to US cities, as the corresponding data set is much more accurate than the UK counterpart.

The direct comparison with Gini and Moran indices present in the original paper might have been misleading, suggesting that the measures are directly comparable, which is not the case since the coverage time measures are based purely on a diffusion process. In the new section, we compared the explanatory power of our framework with widely used segregation measures and a more closely related approach. We keep Gini and Moran for reference. Through this analysis, we show that the spatial diversity measure provides overall better correlation with some socio-economic indicator, and the agreement with Moran I and the sigma index (Ballester and Vorsatz, 2014), which is random walk-based segregation index, indicates that they might be capturing a similar process. We also amended the discussion, pointing to the implications and limitations of our proposal.

We have thoroughly revised the manuscript to further contextualise our results within the current literature on multiscale segregation. As a result, we have also added many references and pointed to several studies which address the problem from a similar angle.

Although we agree with the referee that this kind of work could benefit from multidisciplinary collaborations, we would like to stress that the framework we proposes focuses on the comparison of first-principle quantities with their counterparts computed in meaningful null-models. We believe this approach contributes to better understanding segregation by distinguishing the inherent noise present in the data, together with the strong spatial and cultural constraints of a urban area, from the "signal" pertaining to segregation in a real system. A very positive aspect of the framework we propose is that it also allows to choose a null-model that includes more and more realistic details about the urban area, and thus to measure the deviations from a random pattern due to a specific aspect of interest. The simplicity of the model, in our opinion, is one of the factors that makes it specially appealing to study social dynamics in a urban environment.

=====

Response to Referee 2

=====

We thank the reviewer for the quite positive and encouraging assessment of our work. We have amended the manuscript to address the comments of the referee, especially regarding the necessity to better connect the paper with other existing literature, to include a comparison with other classical and more recent measures of segregation and to better explain our results. More detailed responses inline below.

> The study emphasizes the indices of "spatial variance"
> $\Delta\sigma(c)$ and "spatial diversity" $\Delta\varrho(c)$, which both
> connect to the distribution of the CCT at the node level. For
> example, Figure 3 shows that both $\Delta\sigma(c)$ and
> $\Delta\varrho(c)$ are higher in the cluster-cornered than in the
> cluster-centered case. According to my understanding, any
> segregation measure should order the different cases as follows:
> cluster-centered < (spread-quadrant,spread-corners) <
> cluster-corners. This suggests that the analysis should focus on
> $\mu(c)$.

By switching the focus of our analysis not only to $\mu(c)$, but to the plane with all the three diffusion-based measures introduced in our work, we concluded that the complex spatial patterns governing the synthetic systems can be better explained by the combination of all measures. From this observation, we thoroughly edited the section about synthetic models, separating the interesting case of different patterns with 5 classes.

We included a variety of new numerical simulations, with larger clusters assigned to the "minority" class, which indeed shows a clearer differentiation of the effect of shape and location on coverage times. Although intuitively the simulated segregation regimes might lead the classification suggested by the reviewer -- cluster-centered < (spread-quadrant,spread-corners) < cluster-corners --, we argue that the cluster pattern is in fact the worse configuration for reachability of classes, since leaving a large cluster, on average, requires more steps than smaller ones.

We believe that, in general, the new simulations explain our point in a much clearer manner. In particular, the results consistently show that the shape of a cluster has a more important role than the position of the cluster in determining the class coverage time for walkers started at its nodes. In terms of ethnic segregation, this means that an individual starting from a clustered area would need to visit many more neighbourhoods before meeting new ethnicities, compared to either smaller or more shallow clusters.

A confirmation of our claim comes from the spatial heterogeneity distributions $\mu(c)$ for the example with 32 classes, where shapes are more orderly placed and larger clusters consistently produce larger coverage times.

> Given the vast amount of measures that have been proposed, we have
> reached a point where we have to think beyond a comparison of
> intuitive measures. What characterizes a reasonable segregation
> measure? One way to address this question is by means of an
> axiomatic analysis (normative approach). Another way would be to
> show that particular measures have empirical relevance, that is, one
> should interpret ethnic segregation as a personal characteristic and
> establish that it affects other variables. Finally, one could try to
> derive the analytic properties of the measure (express the measure
> as a function of the matrix that represents the spatial networks).

We have extensively revised the whole manuscript to highlight the relation of the proposed measures with other classical and more recent measures of spatial segregation. We have also included several references to key papers, included those suggested by the reviewer.

Following the suggestion of the reviewer, we included an entirely new section with an in-depth analysis of a variety of other widely used segregation measures, to show how they correlates with some socio-economic indicators usually connected to spatial and social segregation. We compared the explanatory power of our framework with that of those classical indicators. Through this analysis, we show that the spatial diversity measure provides overall stronger correlation with some socio economic indicators, including deprivation, employment, income, and security. The qualitative agreement of these results with the consistent (but relatively lower) correlations with Moran I and the sigma index (Ballester and Vorsatz, 2014), indicates that our measures are capturing a similar process, whilst providing much stronger associations.

We believe that this analysis puts our framework in a better context and helps to show its utility, and we thank the reviewer for the suggestion since the changes make our results more appealing.

In addition, we address uncertainty in our results by comparing the quantities obtained in the real system with the corresponding null-model. We believe this approach contributes to understanding spatial complexity by better distinguishing noise from signal. If all spatial system would be equivalent to their randomised version, we would be capturing only noise, instead, we found significant deviations from randomness.

> In terms of comparisons with other measures, the natural benchmarks
> are the measures proposed by Echenique and Fryer (2007) and
> Ballester and Vorsatz (2014).

The measure in Ballester and Vorsatz (2014) is indeed the closest method with which the CCT quantities can be directly compared. The measure was included in the new table along with several other segregation indices. The definition of SSI index considers only one ethnicity per node, a class could be assigned to a node probabilistic according to its abundance but results would be difficult to interpret and a comparison with the CCT quantities could be meaningless.

=====

Response to Referee 3

=====

We thank the reviewer for the positive appreciation of our manuscript. We have amended the manuscript to clarify the remaining

concern expressed by the reviewer.

> The actual application to measure ethnicity segregation is interest
> but it is not clear how it stands among the whole ecosystems of
> ethnicity segregation measures. Although the authors address the
> differences between theirs and the classical measures, is not clear
> the underlying mechanism behind these differences. I will suggest
> discussing in more mathematical detail the possible reasons behind
> these differences values among the three measures. Besides that, I
> have no objection to recommending this work for publication.

We thoroughly revised the manuscript to better highlight and explain the relation of the proposed measures with other classical and more recent measures of spatial segregation, thus providing more context for our results. We have also updated the section on synthetic systems, adding new simulations to better highlight the differences between our examples with 5 classes. Through this analysis, we explain the dynamics of the shape and location of the spatial patterns on coverage times, and we argue that the cluster pattern is the worse configuration for reachability of classes. This is due to the fact that leaving a large cluster, on average, requires more steps than leaving a smaller ones.

Following the suggestion of the reviewer, we included an entirely new section with an in-depth analysis of a variety of other widely used segregation measures, to show how they correlates with some socio-economic indicators usually connected to spatial and social segregation. We compared the explanatory power of our framework with that of those classical indicators. Through this analysis, we show that the spatial diversity measure provides overall stronger correlation with some socio economic indicators, including deprivation, employment, income, and security. The qualitative agreement of these results with the consistent (but relatively lower) correlations with Moran I and the sigma index (Ballester and Vorsatz, 2014), indicates that our measures are capturing a similar process, whilst providing much stronger associations.

Reviewers' Comments:

Reviewer #1:

Remarks to the Author:

The authors have thoroughly addressed my (and the other reviewer's) concerns and comments. They have devoted much effort to performing new simulations and new analyses, rewriting some sections, and adding new ones. I can now recommend the manuscript for publication.

Reviewer #4:

Remarks to the Author:

Thank you very much for the opportunity review „Quantifying ethnic segregation in cities through random walks“. This is a methodological paper that aims to propose better measures of segregation that are not characterized by biases of existing measures such as scale effects. This is certainly a smart study, and the authors have made great work in proposing random walk based measures that consider the number of steps needed to be made to visit a certain fraction of all classes in the city. The measures are verified through (a) comparisons with other “traditional” measures of segregation and through (b) correlates of segregation such as the labour market characteristics of the cities. The proposed spatial diversity measure is claimed to be superior to traditional measures based on the higher coefficient values with correlates of segregation it yields in Table 1.

As a social scientist, I am not able to comment on the mathematical soundness of the study. My thoughts should thus be considered inferior compared to reviewers of mathematical and data sciences. Risking revealing my ignorance, I share some of the concerns I have with this great paper.

First, the starting point of the paper is summarized in a conceptual map in Figure 1 that depicts two stylized segregation contexts. Although intuitively clear from first glance, I regret to say that I was not able to fully understand the ways it has been constructed. Conceptually, this map is related to terms such as meeting other ethnicities living in the city and to the classes on of neighbourhoods in the paper. What does meeting with other ethnicities exactly mean? Let's take a simple example of ethnicities A and B living in the city. Most likely, there is at least one member of both A and B residing in each neighbourhood of the city. Hence, in each neighbourhood, there is a certain probability for A to meet B. How is this captured in the measures?

Alternatively, it might be that meeting other ethnicities might not be the right term to use. Rather, getting to a different type of neighbourhoods (or classes as used in the paper) with a different mix of As and Bs is what the proposed measure measures. If so, it leaves open the question of how the classes or neighbourhood types have been created? Are there certain thresholds used for proportions of A and B to classify neighbourhoods into different neighbourhood classes? If so, how this would impact the measure?

Second, the verification of the findings in Table 1 needs more care. If I understand it correctly, the claim of the superiority of the proposed measures (more specifically, spatial diversity measure) over the other measures such as Moran I is based on Table 1. This makes this table a really focal element of the study. However, we know very little about the table. For example, more detail is needed on cities, years and ethnicities studied. Segregation measures are very sensitive to ethnic groups studied and they have changed with time.

The coefficients values for spatial diversity in Table 1 are really high, rising up to 0.8, indicating either highly important and genuinely innovative findings or some underlying problems that stem from the assumptions made in creating these measures. Hence, the reader should be better convinced that the first option is the case. Often the socio-economic variables need to be lagged to yield a „true“ relationship. E.g., first incomes should rise for some, and once it has happened, these higher incomes would lead to higher levels of segregation in housing choice.

Third, the scale is not a problem but part of the phenomenon of segregation. Measuring

segregation at different scales, and comparing these findings is often very informative in understanding how segregation is produced and reproduced. Again, I am sorry for the ignorance, but if I understood correctly, the new proposed measures consider scale effects while the other traditional measures don't. If so, could comparisons be somewhat flawed? In short, (a) a good explanation of why the new measures work so well when considering the high values of correlates of segregation; and (b) lag and (c) scale effects with regard to traditional measures could be presented for convincing the reader in the superiority of the proposed measures.

Point-by-point response to the referee's comments

Response to Referee 1

Comment: The authors have thoroughly addressed my (and the other reviewer's) concerns and comments. They have devoted much effort to performing new simulations and new analyses, rewriting some sections, and adding new ones. I can now recommend the manuscript for publication.

We thank the reviewer for the useful comments and suggestions they provided, and we are glad to learn that the reviewer recommended the publication of the revised paper.

Response to Referee 4

Comment: Thank you very much for the opportunity review "Quantifying ethnic segregation in cities through random walks". This is a methodological paper that aims to propose better measures of segregation that are not characterized by biases of existing measures such as scale effects. This is certainly a smart study, and the authors have made great work in proposing random walk based measures that consider the number of steps needed to be made to visit a certain fraction of all classes in the city. The measures are verified through (a) comparisons with other "traditional" measures of segregation and through (b) correlates of segregation such as the labour market characteristics of the cities. The proposed spatial diversity measure is claimed to be superior to traditional measures based on the higher coefficient values with correlates of segregation it yields in Table 1.

As a social scientist, I am not able to comment on the mathematical soundness of the study. My thoughts should thus be considered inferior compared to reviewers of mathematical and data sciences. Risking revealing my ignorance, I share some of the concerns I have with this great paper.

We wholeheartedly thank the reviewer for their positive and supportive assessment of the manuscript, and for the constructive comments included in their report. As researchers in complexity sciences, we truly believe that when it comes to understanding complex phenomena, there is no subject that can be considered superior to any other, or that has the best or more comprehensive answers. Hence, we disagree with the reviewer when they say that their thoughts should be considered "inferior" to any other ones.

We have carefully considered those comments, exactly as we have addressed all the constructive comments of the other reviewers, irrespective of the field of investigation to which those reviewer might have belonged. We have modified the manuscript accordingly, in order to clarify some of the points that could have led the reader to confusion, and to better explain the applicability of our method in the study and comparison of segregation across different systems.

We believe that the revised manuscript has addressed all those concerns thoroughly and in-depth, and we really hope the reviewer might agree in finding the manuscript worthy to be published in Nature Communications.

Comment: First, the starting point of the paper is summarized in a conceptual map in Figure 1 that depicts two stylized segregation contexts. Although intuitively clear from first glance, I regret to say that I was not able to fully understand the ways it has been constructed. Conceptually, this map is related to terms such as meeting other ethnicities living in the city and to the classes on of neighbourhoods in the paper. What does meeting with other ethnicities exactly mean? Let's take a simple example of ethnicities A and B living in the city. Most likely, there is at least one member of both A and B residing in each neighbourhood of the city. Hence, in each neighbourhood, there is a certain probability for A to meet B. How is this captured in the measures?

Alternatively, it might be that meeting other ethnicities might not be the right term to use. Rather, getting to a different type of neighbourhoods (or classes as used in the paper) with a different mix of As and Bs is what the proposed measure measures. If so, it leaves open the question of how the classes or neighbourhood types have been created? Are there certain thresholds used for proportions of A and B to classify neighbourhoods into different neighbourhood classes? If so, how this would impact the measure?

We agree that the discussion of the conceptual map in Figure 1 was probably unclear, and could have led to confusion. In particular, the example reported in Figure 1 is purely fictitious: it does not represent an actual distribution of ethnicities across wards in London. Rather, it presents two extreme scenarios where ethnicities are very segregated or not segregated at all. Also the choice of "the most abundant ethnicity" as a variable of interest was due to the necessity to keep this simple example illustrative and meaningful, even for readers without a specific knowledge in geography. So in this sense the maps are just functional to explain, in the simplest possible way, how spatial heterogeneity and segregation affects the symbolic dynamics sampled by the random walk on the graph of wards adjacencies. It is framed into a (fictitious) metropolitan map just because urban segregation is the whole aim of the paper.

Obviously, each census tract will contain in general several ethnicities, and this is indeed taken into account in the analysis of real-world metropolitan areas that we perform later on in the paper.

We have substantially rephrased the discussion of Figure 1, making it more clear that it is just an illustrative (yet fictitious) example, meant to be used by the reader to construct a simple mental map of the problem we have at hand, and of the kind of solution provided by the methodology we propose. We have removed any mention of “meeting a certain ethnicity”: we have specified what those fictitious colours represent (potential distribution of the most abundant ethnicities in each census tract) and we have rephrased everything in terms of “the random walker visiting a tract of colour X or Y”. We believe that the amended text fully addresses the concerns of the reviewer.

Comment: Second, the verification of the findings in Table 1 needs more care. If I understand it correctly, the claim of the superiority of the proposed measures (more specifically, spatial diversity measure) over the other measures such as Moran I is based on Table 1. This makes this table a really focal element of the study. However, we know very little about the table. For example, more detail is needed on cities, years and ethnicities studied. Segregation measures are very sensitive to ethnic groups studied and they have changed with time.

The coefficients values for spatial diversity in Table 1 are really high, rising up to 0.8, indicating either highly important and genuinely innovative findings or some underlying problems that stem from the assumptions made in creating these measures. Hence, the reader should be better convinced that the first option is the case. Often the socio-economic variables need to be lagged to yield a "true" relationship. E.g., first incomes should rise for some, and once it has happened, these higher incomes would lead to higher levels of segregation in housing choice.

All the information about time and ethnic groups studied is now reported in the “Ethnicity data” section under “Methods”. On top of that, the actual data used to perform the computation is available in a git repository, as indicated in the “Data Availability” statement, at the end of the paper.

Regarding the necessity for the variables to be “lagged”, we argue that the aim of the paper is not to discover the origin and mechanisms which caused a certain level of segregation to appear in a specific metropolitan area. The main focus of the paper is to show that this methodology can be used to capture the essence of segregated patterns, across systems of different size and type, in a consistent manner. Table 1 shows that the levels of segregation measured using our method correlate more strongly with social deprivation indices.

We would like to stress here that there is no specific assumption that might have skewed the results in a direction or another. Actually, Table 1 was added to the paper during the first revision, to answer a question of referee 1 about the actual applicability of these measures as a proxy of some socio-economic indicators usually associated with segregation. We were as surprised as the referee in finding that our measures (initially developed only to quantify spatial correlation in any system, not just ethnic segregation in urban areas) actually end up providing much better proxies of social deprivation. We did not have this result “already in mind” at any point during our research, and there is nothing special about the methodology we propose that was tuned or adjusted in order to obtain those high correlations. Rather, our approach moved from a simple first principle: walks on a coloured graph maintain information about the overall presence of heterogeneity and correlations in the space of colours. We have used the same approach to quantify spatial correlations in many other spatial systems, including cancer tissues and cellular networks of plants, with similar astonishing results (these papers are currently under review)

We must conclude that, indeed, these measures capture something that other measures cannot capture, by the virtue of being based on a simple idea of spatial correlation. We believe that the important ingredients here are the usage of the Class Coverage Times (CCT) and the correct normalisation of CCT values by those observed in the underlying null model, which is what allows us to compare different systems on equal grounds.

We have stressed these facts more strongly in the amended discussion of Table 1, and we have strengthened

the message in the Discussion as well.

Comment: Third, the scale is not a problem but part of the phenomenon of segregation. Measuring segregation at different scales, and comparing these findings is often very informative in understanding how segregation is produced and reproduced. Again, I am sorry for the ignorance, but if I understood correctly, the new proposed measures consider scale effects while the other traditional measures don't. If so, could comparisons be somewhat flawed? In short, (a) a good explanation of why the new measures work so well when considering the high values of correlates of segregation; and (b) lag and (c) scale effects with regard to traditional measures could be presented for convincing the reader in the superiority of the proposed measures.

We believe there is a misunderstanding here, as there are indeed two scale effects to take into account. The first one is the scale of typical tract size, population granularity, and cluster size, while the second one is the scale of the system under study. The first aspect is definitely central to segregation: if we consider the entire metropolitan area as one big tract, there is no way to define a segregation level for it. But unfortunately, this kind of scale problems are mainly due by the availability of data, or on the focus of the study: we can definitely say that a city looks more or less segregated at the level of wards rather than at the level of entire boroughs, and vice versa, because we are comparing the same city at two different scales. But it would make little sense to compare the segregation of London at the level of wards with the segregation in Leeds at the level of boroughs!

In this sense, the second aspect of segregation (different system sizes) is instead meaningful when we compare different cities, possibly at different scales, and effectively becomes a potential confounding factor. If we consider census tracts of the same size/population density in a large city as London and in a smaller city as Leeds, then the results obtained by most measures will be size-dependent, due to the fact that London has a larger number of tracts, and most of the classical measures of segregation actually are size-dependent in this sense. By using the Class Coverage Times, and normalising them by the corresponding values observed in the null-model, our method washes out most of that size-dependency, and allows us to compare London and Leeds (but also London and Houston) on equal grounds.

We have added an entire new paragraph in the Discussion to explain this fundamental aspect. Moreover, we have also included a new study as Supplementary Figure 3 that takes into account the same checkboard-like arrangement of clusters of four colour on a 2D lattice, with different typical size of clusters. This figure confirms that the measures we propose are sensible to the actual typical size of each cluster, which is undoubtedly an important facet of segregation.

Reviewers' Comments:

Reviewer #4:

Remarks to the Author:

Thank you very much for the revised version of the paper. I am happy with the response and I do not have any further comments.